# Successor Options : An Option Discovery Algorithm for Reinforcement Learning

## Abstract

Hierarchical Reinforcement Learning is a popular method to exploit temporal abstractions in order to tackle the curse of dimensionality. The options framework is one such hierarchical framework that models the notion of skills or options. However, learning a collection of task-agnostic transferable skills is a challenging task. Option discovery typically entails using heuristics, the majority of which revolve around discovering bottleneck states. In this work, we adopt a method complementary to the idea of discovering bottlenecks. Instead, we attempt to discover "landmark" sub-goals which are prototypical states of well connected regions. These sub-goals are points from which a densely connected set of states are easily accessible. We propose a new model called Successor options that leverages Successor representations to achieve the same. We also design a novel pseudo-reward for learning the intra-option policies. Additionally, we describe an Incremental Successor options model that iteratively builds options and explores in environments where exploration through primitive actions is inadequate to form the Successor representations. Finally, we demonstrate the efficacy of our approach on a collection of grid worlds and on complex high dimensional environments like Deepmind-Lab.

## 1 Introduction

Reinforcement Learning (RL) (Sutton & Barto, 1998) has garnered significant attention recently due to its success in challenging high-dimensional tasks (Mnih et al., 2015; 2016; Silver et al., 2016; Hessel et al., 2017; Lillicrap et al., 2015; Oh et al., 2016; Schulman et al., 2015). Deep Learning has had a major role in the achievements of RL by enabling generalization across a large number of states using powerful function approximators. Nevertheless, there exists vast room for improvement in terms of exploration and sample complexity that cannot be addressed solely using Deep Learning. Hierarchical Reinforcement Learning (HRL) is one potential strategy that mitigates the curse of dimensionality using temporal abstraction. Recent work (Vezhnevets et al., 2017; Kulkarni et al., 2016a; Bacon et al., 2017) has attempted to use a hierarchy of controllers operating in different time-scales, in order to search large state spaces rapidly.

The options framework (Sutton et al., 1999) is an example of a hierarchical approach that models temporally extended actions or *skills*. The discovery of options has not been effectively addressed and is known to be a hard task. Meticulously designing the options is often required to reap the benefits of the options framework. While there are a number of approaches to this problem, a large fraction of literature revolves around discovering options that navigate to *bottleneck states* (McGovern & Barto, 2001; Şimşek & Barto, 2004; Şimşek et al., 2005; Menache et al., 2002). In this work, we adopt a paradigm that fundamentally differs from the idea of identifying bottleneck states as sub-goals. Instead, we attempt to discover *landmark* or *prototypical states* of well connected regions while ensuring that they are also spread apart. Building on this intuition, we propose *Successor Options*, a sub-goal discovery and intra-option policy learning framework.

While bottlenecks have their utility in many scenarios (discovering keys or doors), Successor options discover options that are likely to be better for exploration. The sub-goals provide greater accessibility to a larger number of states within a well connected region allowing an agent to effectively explore within that region. We validate this claim on a number of grid world tasks and observe faster learning in all of the tasks. To discover the options, we leverage Successor representations

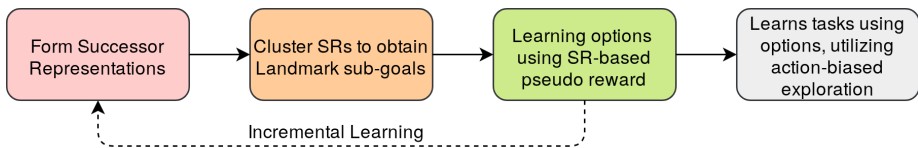

Figure 1: Overview of steps involved in our option discovery algorithm

(SR) (Dayan, 1993) to learn the sub-goals and the intra-option policy. Successor representation is a vector encoding the expected discounted visitation counts of all states in the future. The SR varies with the policy since the visitation counts of the states depend on the policy being executed to estimate these counts. The first step in our option discovery method involves collecting the SRs of all states. The sub-goals are then identified by clustering a large sample (or all) of the SR vectors and assigning the cluster centers as the various sub-goals. This steps discovers a collection of goals which are well separated in the Successor representation space. Since the SR encodes information about the consequently occurring states, the cluster centers translate to sub-goals that have vastly different successor states, meaning different sub-goals provide access to a different region in state-space. The overview of our option discovery procedure is summarized in Figure 1.

Building the Successor representation however relies on the usage of primitive actions to navigate the state space (required to estimate discounted visitation counts). To mitigate the same, we propose an *Incremental Successor Option* algorithm. This method works in an iterative fashion where each iteration involves an option discovery step and an SR update step. The updated SR is used to form a more accurate set of options. These newly formed options improve the SR updates in the consequent iteration. This synergy enables the rapid learning of Successor representations by improving sample efficiency.

Successor Options offer a number of advantages. While an intermediate clustering step segments the algorithm into distinct stages (non-differentiability introduced), the step is critical in many aspects. Firstly, the number of options $k$, is specified beforehand which allows the model to adapt by finding $k$ most suited sub-goals. The algorithm does not require the pruning of options from a very large set unlike other works (McGovern & Barto, 2001; Şimşek & Barto, 2009; Machado et al., 2017). The clustering also ensures robust options since the cluster centroids are an average of many Successor representations. Furthermore, the discovered options are reward agnostic and are easily transferable across multiple tasks. This enables solving a plethora of tasks varying solely in reward structure, in a rapid fashion. The sub-goals are relevant even in cases where the environment has no natural bottlenecks and hence has wider applicability. Lastly, the algorithm is easily extended to function approximators since Successor representations have a suitable neural network formulation (Kulkarni et al., 2016b; Barreto et al., 2017).

The principle contributions of this paper are as follows : **(i)** An automatic option discovery mechanism revolving around identifying landmark states, **(ii)** A novel pseudo reward for learning the intra-option policies that extends to function approximators **(iii)** An incremental approach that alternates between exploration and option construction to navigate the state space in tasks with a fixed horizon setup where primitive actions fail to explore fully.

## 2 PRELIMINARIES

Reinforcement Learning deals with sequential decision making tasks and considers the interaction of an agent with an environment. It is traditionally modeled by a Markov Decision Process (MDP) (Puterman, 1994), defined by the tuple $\langle \mathcal{S}, \mathcal{A}, \mathcal{P}, \rho_\iota, r, \gamma \rangle$, where $\mathcal{S}$ defines the set of states, $\mathcal{A}$ the set of actions, $\mathcal{P} : \mathcal{S} \times \mathcal{A} \rightarrow \mathcal{S}$ the transition function, $\rho_\iota$ the probability distribution over initial states, $r: \mathcal{S} \times \mathcal{S}' \times \mathcal{A} \rightarrow \mathcal{R}$ the reward function and $\gamma$ the discount factor. A policy dictates the behaviour of the agent in an environment. More formally, a policy is denoted by $\pi(s): \mathcal{S} \rightarrow \mathcal{P}(\mathcal{A})$, where $\mathcal{P}(\mathcal{A})$ defines a probability distribution over actions $a \, \epsilon \, \mathcal{A}$ in a state $s \, \epsilon \, \mathcal{S}$. In the context of optimal control, the objective is to learn a policy that maximizes the discounted return $R_t = \sum_{i=t}^{T} \gamma^{(i-t)} \, r(s_i, s_{i+1}, a_i)$, where $r(s_i, s_{i+1}, a_i)$ is the reward function.

**Deep Q-learning:** Q-learning (Watkins & Dayan, 1992) attempts to estimate the optimal action-value function $Q^*(s, a)$. It exploits the Bellman optimality equation, the repeated application of which leads to convergence to $Q^*(s, a)$. The optimal value function can be used to behave optimally by selecting action $a$ in every state such that $a \in \text{argmax}_{a'} Q(s, a')$

$$Q(s_t, a_t) \leftarrow Q(s_t, a_t) + \alpha \left[ r_{t+1} + \gamma \max_{a'} Q(s_{t+1}, a') - Q(s_t, a_t) \right] \tag{1}$$

Mnih et al. (2015) introduce Deep Q-learning, that extends Q-learning to high dimensional spaces by using a neural network to model $Q_\theta(s, a)$. The authors use the TD-error to back-propagate the gradients which are used to learn the parameters $\theta$. A delayed target-network and an experience replay (Mnih et al., 2015) are typically utilized to stabilize the training and reduce the correlation between samples in a mini-batch.

**Options and Semi-Markov Decision Processes:** Options (Sutton et al., 1999) provide a framework to model temporally extended actions. Formally, an option is defined using the 3-state tuple : $\langle \mathcal{I}, \beta, \pi \rangle$, where $\mathcal{I} \subseteq \mathcal{S}$ is the initiation set, $\beta : \mathcal{S} \to [0, 1]$ the termination probabilities for each state and $\pi : \mathcal{S} \to P(\mathcal{A})$ the intra-option policy. In this work, we assume that the intra-option policies satisfy the Markov assumption.

The options can be understood as a sequence of actions and hence naturally fit into the Semi-Markov decision process framework (Puterman, 1994). The Markov nature of intra-option policies can be exploited using intra-option value function updates (Sutton et al., 1999). This enables learning the value function associated with one option while executing another.

**Successor Representation:** The Successor Representation (SR) (Dayan, 1993) represents a state $s$ in terms of its successors. The SR for $s$ is defined as a vector of size $|\mathcal{S}|$ with the $i^{th}$ index equal to the discounted future occupancy for state $s_i$ given the agent starts from $s$. Since the SR captures the visitation of successor states, it is directly dependent on the policy $\pi$ and the transition dynamics $p(s_{t+1}|s_t, a_t)$. More concretely, the SR can be written as follows:

$$\psi_\pi(s, s') = \mathbb{E}_{s' \sim P, a \sim \pi} \left[ \sum_{t=0}^{\infty} \gamma^t \mathbb{I}(s_t = s') \mid s_0 = s \right] \tag{2}$$

where, $\mathbb{I}(.)$ is 1 if its argument is true, else 0 (indicator function). The SR can be learnt in a temporal difference (TD) like fashion by writing it in terms of the SR of the next state.

$$\hat{\psi}(s, :) \leftarrow \hat{\psi}(s, :) + \alpha \left[ \mathbb{1}_s + \gamma [\hat{\psi}(s', :)] - \hat{\psi}(s, :) \right] \tag{3}$$

The above equation is for state samples $s, s'$, where $\hat{\psi}$ is the estimate of SR being learnt and $\mathbb{1}_s$ is a one-hot vector with all zeros except a 1 at the $s^{th}$ position. Successor Representations can be naturally extended to the deep setting (Kulkarni et al., 2016b; Barreto et al., 2017) as follows (note $\phi(s)$ is feature representation of $s$) :

$$\psi_\pi(s_t; \theta) = \mathbb{E} \left[ \phi(s_t) + \gamma \, \psi_\pi(s_{t+1}; \theta) \right] \tag{4}$$

## 3 PROPOSED METHOD

Option discovery has typically involved some form of bottleneck discovery in many prior works. Bottleneck states can be discovered in a myriad of ways, such as finding states that are frequently visited while executing successful trajectories (McGovern & Barto, 2001), or by identifying states that connect different densely connected regions of the state space (Menache et al., 2002). Learning options that go to such bottleneck states have been shown to accelerate learning and improve exploration. Successor Options (SR-options) adopts an approach which attempts to discover sub-goals that are representative of well connected regions. This technique exhibits two advantages, **(i)** of learning useful options without extrinsic reward, i.e. latent learning scenarios and **(ii)** of learning to explore the state space incrementally in scenarios where primitive actions are unable to facilitate full exploration.

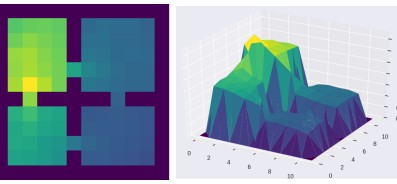

Figure 2: Successor Representations in two room domain: The first two images from the left are the SRs for states in two different rooms. The last two images show SR as pseudo reward used for learning an option that terminates at one of the cluster centers (shown in yellow).

### 3.1 SUCCESSOR OPTIONS

In learning Successor Options, the first step involves learning the SR. This is followed by clustering states based on the learnt SR (We utilize K-means++ (Arthur & Vassilvitskii, 2007)). Since the SR captures temporally close-by states efficiently, we are able to generate clusters which are spread across the state space, with each cluster assigned to a set of states that are densely connected. The SR can be understood to encode the connectivity of the underlying graph of the environment as evident in Figure 2. We wish to learn options over a given clustering, since the cluster centers act as landmark states or sub-goals for these options. Hence, a sub-goal is that state whose SR has the largest cosine similarity to a corresponding cluster center. The sub-goal discovery step is followed by learning the intra-option policies corresponding to every sub-goal.

**Latent Learning:** In order to learn option policies, we attempt to ascend the SR of the cluster center. This intuition is translated into an appropriate reward over which the policy is learnt. Figure 2 is the Successor Representation of a state learnt using the uniformly random policy which highlights why climbing the SR of the sub-goal is a valid choice. The reward function used, is formalized in equation 5 ($\psi_s$ is the SR of state $s$) . This provides a dense reward which ensures that steps toward the sub-goal are positively rewarded while moving away from the cluster center is penalized. Note that an approximately developed SR is often a sufficient signal for learning optimal options.

$$r_{\psi_s}(s_t, a_t, s_{t+1}) = \psi_s(s_{t+1}) - \psi_s(s_t) \tag{5}$$

Every option has every state in it's initiation set. The termination set of every option is the set of all states satisfying $Q(s, a) \leq 0$. The intra-option policy, is that learnt from the pseudo reward. Now that the options are obtained, the only remaining step is to use SMDP Q-learning to learn a policy over the discovered options.

### 3.2 INCREMENTAL SUCCESSOR OPTIONS

Here we consider the case where the agent is unable to explore the whole state space with only primitive actions in finite time. This hinders the learning of usable Successor representations, critical to the SR options algorithm. We wish to explore incrementally and learn options which cover all parts of the state space in order to facilitate learning for reward based tasks. This is especially important in sparse reward long horizon tasks where exploration can be facilitated using reward independent options. While incrementally exploring and learning options, no extrinsic reward is considered. We propose iteratively learning the SR and options using the above mentioned clustering scheme. The policy used for learning the SR is augmented with the previously learnt options to allow traversing between distinct clusters in the state space. However, since the SR directly depends on the policy, we do not update the SR while executing an option. This is desirable as we want to learn SR for a random policy and not for an option biased policy. Executing the options allows the agent to land in well spread parts of the state space around which learning takes place only through random actions. Therefore, the augmented policy only uses options to quickly reach dissimilar parts of the state space and not for learning the SR itself.

---

**Algorithm 1 :** Incremental Successor Option Learning

---

$A \leftarrow \{a_0, a_1, a_2, ..., \}, a_i \, \epsilon \, \mathcal{A}, O \leftarrow \{\}$      // Starting with only primitive actions
**for** $i = 1, ..., N$ iterations **do**
    $\psi \leftarrow \text{LearnSR}(O + A)$      // Eq. 3
    $C \leftarrow \text{GetCandidateSubgoals}()$
    $S \leftarrow \text{ClusterSR}(C, n)$
    Empty option set $O \leftarrow \{\}$      // Relearning $n$ new options every step
    **for each** sub-goal $s \, \epsilon \, S$ **do**
        $o \leftarrow \text{LearnOption}(\psi(s))$      // Q learning with SR pseudo-reward
        Store $o$ in O
    **end for**
**end for**
**return** $S$, O

---

In the above described algorithm, candidate sub-goals are a fraction of reached states. Formally, a state $s$ is a candidate sub-goal if $SR_{min} < \sum_{s'} \psi(s, :) < SR_{max}$, where $SR_{min}$ and $SR_{max}$ are the first and third quartile values of SR sums of all reached states respectively. Such a condition ensures that all candidate sub-goal states have an SR that is neither fully developed nor extremely sparse or under developed, thus providing a pseudo reward which is easy to learn over. This also allows candidate states to be states which lead to more exploration since these are not visited as frequently as some of the others.

### 3.3 SUCCESSOR OPTIONS IN THE FUNCTION APPROXIMATION SETTING

In this section we extend our formulation for successor options to real world tasks with large number of achievable states. Kulkarni et al. (2016b); Barreto et al. (2017) propose an architecture for generating the SR in the function approximation case or Successor Feature (SF) by learning from three signal branches, i.e. the reward prediction error, image reconstruction error and the TD error for learning Successor Features. All three branches share the same base representation $\phi(s)$ for learning here. The reward prediction is required in order to compute the Q values as the dot product of the SF $\psi(s)$ and reward weight vector $w$. Since we do not learn for control in our case, we do not include the reward layer in our architecture. Unlike other works (Machado et al., 2017; Lakshminarayanan et al., 2016) we do not need to approximate or compute a graph Laplacian and the formulation can be naturally extended to neural networks. The architecture used, is presented in Figure 4. Generally, an image reconstruction task acts as an auxiliary task to stabilize training and avoid learning a trivial SF representation. We use a similar task of reconstructing an intermediate representation instead of predicting the input image itself. The auxiliary loss hence takes the form $\mathcal{L}_{auxiliary} = $ Error in Reconstruction of layer.
Following the training of SF, we collect sufficient samples of the SF vectors and stack them to form our sample SF matrix. Similar to the tabular case, this matrix is clustered to produce SF cluster center. For learning corresponding options, we use the dense reward given by the cluster center SF ($\psi_s$) value as the reward in a standard DQN training procedure. The weights for the shared representation are frozen when learning option policies. This is required since the rewards depend on $\phi(s)$ and hence should not change for a particular state. Equation 6 gives the reward function($\psi_s$ is the SR of one of the cluster centers) used to train the intra-option policy.

$$r_{\psi_s}(s_t, a_t, s_{t+1}) = \psi_s \cdot (\phi(s_{t+1}) - \phi(s_t)) \tag{6}$$

## 4 EXPERIMENTS AND RESULTS

We demonstrate our method on four different grid world tasks for the tabular setting and on the Deepmind-Lab Suite tasks for the function approximation case. The grid world tasks (Figure 3) are designed with increasing complexity both in terms of the number of states available and the structure of rooms. This is done so as to test the generality of Successor options in learning well spread out options, each being representative of states enclosed in close-by rooms.

$$\mathcal{L} = \mathcal{L}_{SR} + \mathcal{L}_{auxiliary} \tag{7}$$

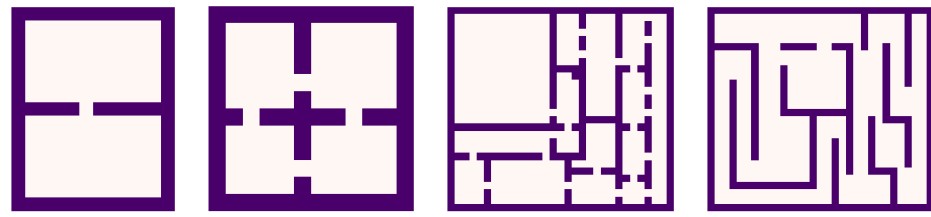

Figure 3: Grid world Tasks

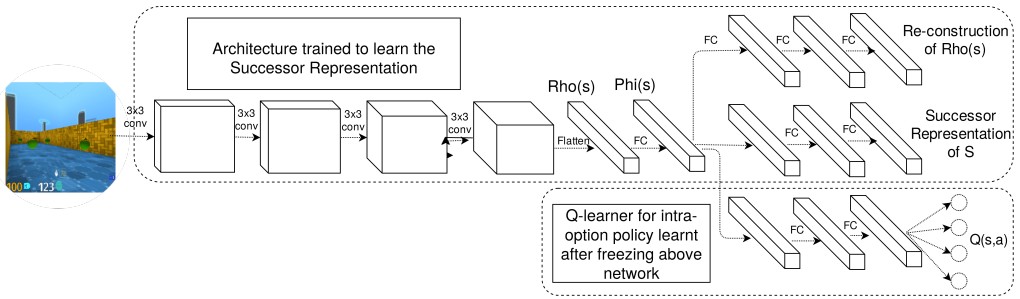

Figure 4: Neural Network Architecture for Deep Successor Options : The Architecture is trained in two stages. The first step involves learning the SR augmented by the re-construction loss. The previously learnt weights are frozen following which each option learns an optimal action value function for the corresponding pseudo reward

**Visualizing Successor Option Sub-goals** The sub-goals obtained using our clustering scheme are more spread out as compared to those obtained using the eigen-option discovery framework (Machado et al., 2017) as evident from Figure 5. Moreover, it can be easily seen that each sub-goal works as a good landmark state for all states very close to it, i.e. landing in any of the sub-goals allows efficiently exploring other states around them when learning for goal directed behavior. This is not the case with sub-goals for eigen-options, which tend to be more around corners with multiple sub-goals being discovered in the same region.

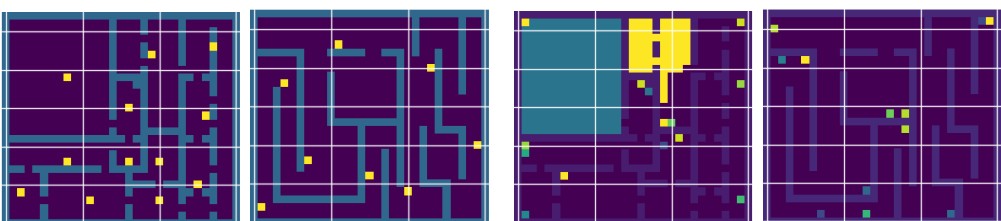

Figure 5: Visualizing sub-goals for successor option (left) and eigen-options (right). Different colour of the sub-goals (on right image) correspond to different eigen-vectors. A single eigen-vector could have multiple sub-goals (ex: any state in yellow region is a sub-goal for that option)

**Successor Options vs Eigen-Options** We compare successor options with eigen-options with respect to performance in learning extrinsic reward based tasks (see Figure 6). During SMDP Q-learning, a uniformly random policy among the options and actions (typically used in exploration) will result in the agent spending a large fraction of it's time near these sub-goals. Machado et al. (2017) also observe this behaviour when working with eigen-options. Hence, in order to effectively explore the environment using the exploration policy, it is important to sample actions and options non-uniformly. Therefore, we utilize variants of both successor and eigen-options called successor non-uniform and eigen non-uniform options which refer to sampling primitive actions at a higher probability than the options(sampling in the exploratory policy). In all our experiments, we fix the ratio of sampling an option to an action as 1:19.

The task used to compare the algorithms was a binary reward task where an agent was expected to navigate to the goal(+1 on reaching goal and +0 otherwise). We use (4, 5, 10, 10) options in each of the grid worlds. 500 different instances of the task were run, each with a random start and goal state and the reported plots were averaged over these 500 instances. In order to gauge the speed at which each algorithm learns, we use the area under the return curve (AUC). Let $N$ be the total number of steps (primitive actions) taken in the environment during training. After every $K$ steps, we test the performance of the agent and note the return. The sum of the $\lfloor \frac{N}{K} \rfloor$ different evaluations (each occurring at an interval of $K$ steps) is defined as the *Area under the Return Curve*.

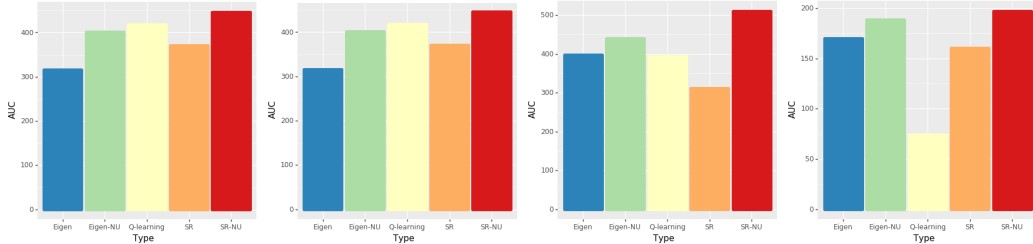

Figure 6: Comparing Successor Options with eigen-options for both uniform and non uniform sampling and with simple Q Learning using just primitive actions.

**Incremental Successor Options** We consider scenarios where learning from only primitive actions is insufficient. Such a case resembles many real world tasks and therefore is important to show the generality of our method. For instance, in a robotic manipulation task of pushing a block, given enough freedom in the setup, the agent fails to hit the block with random primitive actions in given time steps. To realize this distinction, we consider working strictly in a finite horizon setting even when learning without an extrinsic reward, i.e. the agent restarts from a single state after every N number of steps. Since a random policy of primitive actions is not able to reach all states inside the horizon, adding such a constraint now makes learning the SR extremely hard for the grid world tasks we consider. We show how using the Incremental Successor options, we are able to explore intelligently, gradually learning the SR and finding sub-goals that slowly spread out across the whole state space (see Figure 7).

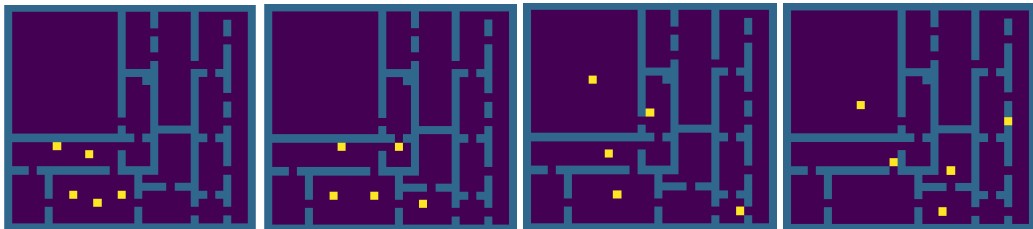

Figure 7: Incremental Successor Options : We are able to learn spread out sub-goal policies with an SR reward after a few iterations (left to right) where as naively iterating using SR as a proxy for visitation count does not facilitate such exploration.

**Deep Successor Options** We show how clustering the successor feature vectors after learning the deep successor representation can allow learning useful options in the maze task of the Deepmind-Lab Suite (Figure 8). For learning option policies, we use standard DQN with the reward given as in equation 6. We train a total of 8 options. We noted that one option crosses the doorway to get to the other room. Another option learns to walk to the end of a corridor from the central part of the same. The t-SNE visualization presented in the same figure indicates the learnt SR is able to categorize the states into discrete categories. This is evident from the separation among groups of data-points.

## 5 RELATED WORK

Moore et al. (1999) introduce airport hierarchies which assign different states as airports or landmarks with various levels defined on the basis of seniority. Each state is assigned to be a landmark only if it is reachable from a threshold number of states. The airport analogy is similar to the spread

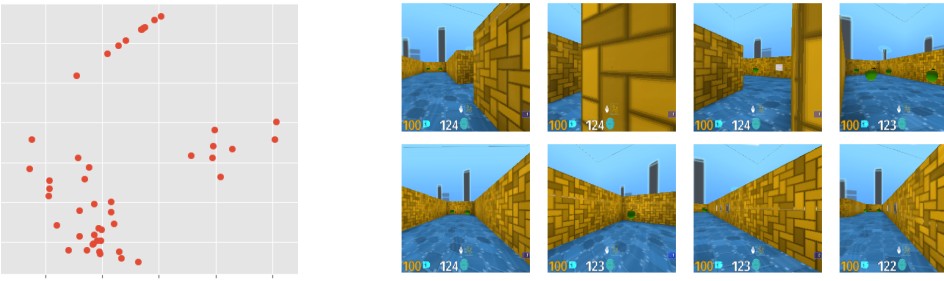

(a) t-SNE visualization of the SF for maze task

(b) Visualizing two of the learnt options in the Deepmind Lab Maze task

Figure 8: Deep Successor Options for Maze task

of clusters that we obtain for discovering options, since each airport also represents a group of similar states. Moreover, while learning iteratively, since we only use states with an adequately learnt SR , the candidate cluster centers are always able to leverage the SR as a dense reward and in turn learn successful option policies.

The overwhelming majority of literature on option discovery employ techniques that revolve around discovering bottleneck states. McGovern & Barto (2001) describe a diverse density based solution that casts this problem as a multiple-instance learning task. The discovered solutions are bottlenecks since they are present in a larger fraction of positive bags. Şimşek & Barto (2009) describe a betweenness centrality based approach which also naturally lead to bottleneck based options. Sub-goals based on relative novelty (Şimşek & Barto, 2004) identify states that could lead to vastly different states consequently which is closely tied to the notion of bottleneck states.

Graph partitioning methods have also been employed to find options (Şimşek et al., 2005; Lakshminarayanan et al., 2016; Menache et al., 2002). These methods design options that transition from one well connected region to another. Since the sub-goals are the boundaries between two well connected regions, these method also typically identify bottlenecks as sub-goals. In contrast, our work attempts to discover sub-goals that are representative of a well-connected region and are well separated in addition.

Option Critic (Bacon et al., 2017) is an end-to-end differentiable model that learns options on a single task. However, this method is forced to specialize to a single task and the learnt options are not easily transferable. Eigen-options Machado et al. (2017) use the eigen-vectors of the Laplacian as rewards to learn intra-option policies. This method however lacks a variety of sub-goals since ascending the different eigen-vectors often correspond to reaching the same sub-goal. Our proposed model uses a clustering stage to ensure that the sub-goals are sufficiently varied in nature. The clustering step also provides flexibility regarding the number of options required, which is absent in the case of eigen-options.

# 6 CONCLUSION

We propose a scheme for discovering options based on clustering the Successor representations. We demonstrate that this method produces well separated and suitably located sub-goals which allows faster exploration when learning for extrinsic reward based tasks. We also propose an iterative approach to this setting where primitive actions are unsuitable for navigating the state space. We show how using such an approach in latent learning scenarios allows exploring the environment incrementally and yields useful options that can be reused for reward based tasks as well.

As future work, we aim to use Deep Successor options to achieve optimal control on sparse reward tasks. This work assumes that the initiation set is the set of all states. We plan on looking into ways in which we can exploit the connectivity information of the SR to determine the initiation set. Currently, our setup uses very strict termination conditions which we could potentially turn into a soft probabilistic approach. Our current approach does not use the rewards to shape the nature of the options. We plan on looking into techniques that alter the policy used to generate the Successor representation based on extrinsic rewards to build SRs on relevant parts of the state space.

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

APPENDICES

## A    DEEP SUCCESSOR OPTION LEARNING ALGORITHM

---
**Algorithm 1 :** Deep Successor Option Learning

---
$\psi \leftarrow$ LearnSF()                                                           // Eq. 4
**for** $t = 1, ..., H$ time steps  **do**
    Store $\psi(s_t)$ in T                     // Using random policy of primitive actions
**end for**
 S $\leftarrow$ ClusterSF(T)
**for each** subgoal $\psi_c \, \epsilon \, S$ **do**
    $o_c \leftarrow$ LearnOption($\psi_c$)                  // DQN with reward from equation 6
**end for**

---

## B    GRID WORLD DESCRIPTION

We have 4 grid worlds in total. The dimensions of the grid worlds are $13 \times 10$, $10 \times 10$, $30 \times 23$ and $26 \times 30$ (left to right in Fig 3). The first two grid worlds are the standard two-room and four-room domains. The tasks used for evaluation are navigation tasks where the aim is to reach a certain randomly selected goal from a random starting location (500 such randomly chosen scenarios were considered). The value of $\gamma = 0.95$ for each of these tasks so that the agent is encouraged to finish the task in least possible time. The agents have 5 actions from every state which include Left, right, forward, backward and No-op. The task has a reward of 0 for every transition barring the transitions into the goal state which have a reward of +1.

## C    DEEPMIND-LAB DOMAIN

The Deepmind-Lab task is a partially observable MDP and is a maze navigation task. We designed our own custom maps to test the Deep Successor options Algorithm. The 4 available actions are rotate left, rotate right, move forward or move backward. The input image is of size $320 \times 240$ pixels(RGB) and $\gamma = 0.99$.
We designed a task presented in Table 1. * indicates walls, $P$ indicates potential player spawn point, $G$ indicates the Goal (+10 reward) and A indicates Apple (+1) reward. (notation consistent with Deepmind-Lab lua Map API)

| * | * | * | * | * | * | * | * | * | * | * | * | * | * |
|---|---|---|---|---|---|---|---|---|---|---|---|---|---|
| * | G | P | A | P | A | P | P | P | P | P | P | P | * |
| * | P | A | P | A | P | P | P | P | P | P | P | P | * |
| * | * | * | * | * | * | P | * | * | * | * | * | * | * |
| * | P | P | P | P | P | P | P | P | P | A | P | A | * |
| * | A | A | P | P | P | P | P | A | P | P | A | P | * |
| * | * | * | * | * | * | * | * | * | * | * | * | * | * |

Table 1: DeepmindLab Task

## D    SUCCESSOR REPRESENTATION AND STATE GRAPHS

Successor Representations encode the information regarding the state connectivity without explicitly modelling the same. This is advantageous over other methods that work with the graph Laplacian since building graphs become intractable in very large state spaces. To understand the same, we analyze the 2-dimensional projections of SRs obtained from grid worlds using t-SNE (Maaten & Hinton, 2008). Figure 9 clearly indicates that the adjacency information of the states are present in the SR matrix. The number of distinct clusters are highly indicative of the number distinct well connected regions present in the environment.

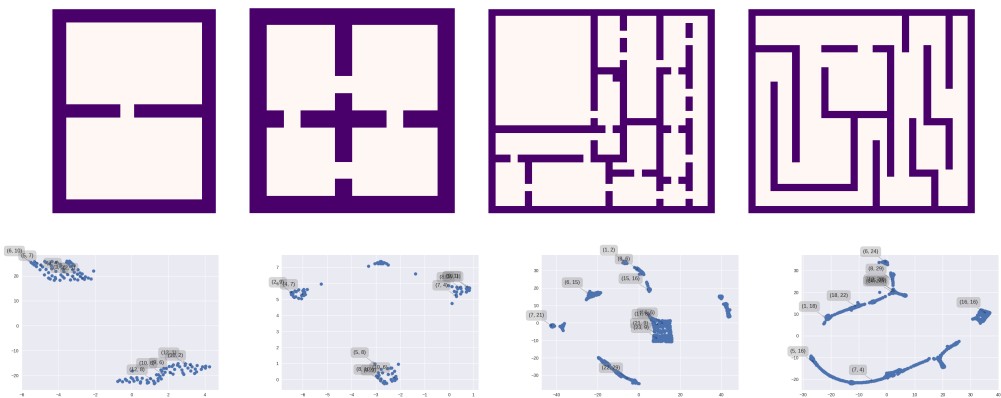

Figure 9: Visualizing successor representations using t-SNE

# E  COMPARING CLUSTERING TECHNIQUES AND METHODS

The clustering of Successor representations involve a number of different decisions. The clustering algorithm is one such choice we experimented with. We looked at three algorithms, namely K-means++ (Arthur & Vassilvitskii, 2007), K-medoids (Park & Jun, 2009) and a greedy approach. The greedy approach is an iterative approach where each iteration adds a new sub-goal to an existing list. The added sub-goal is selected based on the criteria of having the maximum value for the minimum distance to all other existing list of sub-goals. The algorithm is a simple $\mathcal{O}(n^2)$ routine. We use the same evaluation procedure as that used Section 4, in order to compare eigen-options and Successor options. We also run a uniform and non-uniform option-action sampling procedure for each clustering method. The performances are shown in Figure 10. The plots indicate that using the K-means++ routine to obtain centroids (which is followed by choosing states closest to each of these centroids as sub-goals) yields the highest performance.

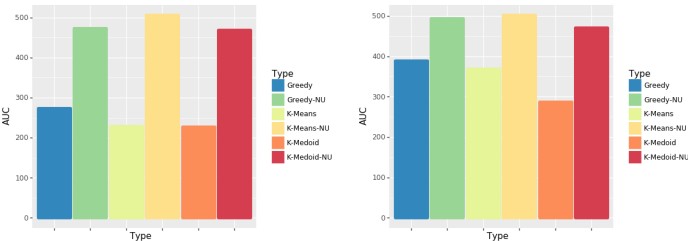

Figure 10: Comparison of Clustering Techniques (for 3rd (left) and 4th (right) environments)

Euclidean distance based clustering algorithms like K-means also rely on normalizing (normalization occurs across features) the various data-points across dimensions. We look at the effect of the same on performance, using a setup identical to that in Section 4, which was used to compare eigen-options and Successor options. From Figure 11, the unnormalized version of the SR has a superior performance. We attribute this to the fact that the $L_1$ norm of the SR is identical for every state on convergence. Hence, there is no explicit need for a normalization step.

The equation below gives the expression for the $L_1$ norm of the SR. It is in-fact dependent only on $\gamma$ as evident from the equation.

$$|\psi(s)|_1 = 1 + \gamma + \gamma^2 + \cdots$$
$$= \frac{1}{1 - \gamma}$$

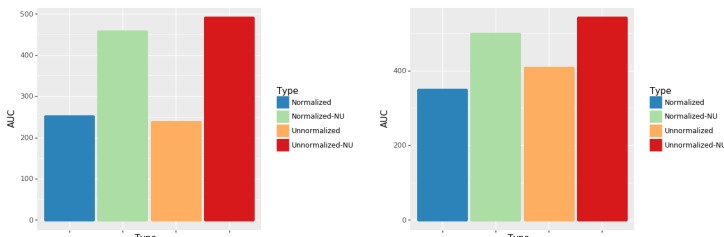

Figure 11: Comparison of Normalization Techniques(for 3rd(left) and 4th(right) environments

## F   INCREMENTAL SUCCESSOR OPTIONS

In Figure 12, we show how the sub-goals become increasingly spread out as they are learnt iteratively using Incremental Successor options. We use 5 options in both grid worlds and the horizon is set to 100 time steps. We observe that learning naively by selecting the rarest states as sub-goals does not help in exploration. This is because the SR for such states is extremely sparse and therefore does not provide a rich signal to learn options from. Moreover, since such least visited states are present close-by, the sub-goals selected are not evenly spread out.



Figure 12: Sub-goals discovered for grid world 4 for different iterations (left to right)

## G   EXPERIMENTAL DETAILS - GRID WORLD

The grid world tasks learnt (4, 5, 10, 10) options on the corresponding environment. For the case of eigen-options, an adjacency graph over the states of the environment was built by the agent. The eigen-vectors of the graph Laplacian are then obtained. These eigen-vectors are used as rewards for learning the eigen-options(as dictated by Machado et al. (2017)). The positive and the negative of the same eigen-vector were used to generate options and were counted as separate options. The Successor representations were collected using the uniformly random policy run for $5 \times 10^6$ steps in the environment. The SRs were then clustered using the K-means algorithm following which a Q-learner learns the option policy in $1 \times 10^6$ steps.

Once both the Successor options and Eigen-options for an environment were discovered, they were used in an SMDP Q-learning framework to learn an option policy. A non-uniform sampling of option to actions (1:19) was attempted instead of the uniformly random sampling for the exploratory action. The algorithms were trained for $5 \times 10^5$ steps (primitive actions) in the environment. A $\gamma = 0.99$ was used for this task. The SMDP Q-learning updates were augmented with intra-option action-Value function updates presented by Sutton et al. (1999). These updates exploit the fact that intra-option policies could take the same action from the same state. In such a scenario, the value functions of both options can be updated.

## H   EXPERIMENTAL DETAILS - DEEPMIND-LAB

The Deepmind-Lab task was trained using DQN (with experience replay and target network). Double Q-learning was also used to stabilize training. The value of $\epsilon$ was annealed from 1.0 to 0.1 in 1 Million steps. A batch size of 32 was used with $\gamma = 0.99$. The same learning rate of $2.5 \times 10^{-4}$ was used across all stages. The training of the SR occurred for 5 Million steps and each option policy was trained for 2 Million steps. A memory of size 100,000 was used. The target network was updated every 10,000 steps. RMS-prop was used as the gradient optimizer technique.

| Base layer |
|:---:|
| 8 x 8 conv, 8 x 8 conv, 4 x 4 conv, 4 x 4 conv, 3 x 3 conv, 512 fully connected, 128 fully connected |
| **Reconstruction branch** |
| 128, 128 fully connected |
| **SR branch** |
| 64, 64, 128 fully connected |

## I PERFORMANCE CURVES

We present the performance curves as a function of steps in the environment in Figure 13. These are the same curves whose area is reported in Figure 6. It is clear to see that SR-options result in better returns throughout training. The plot clearly highlights the gulf in performance between the various methods in comparison to our method SR-options.

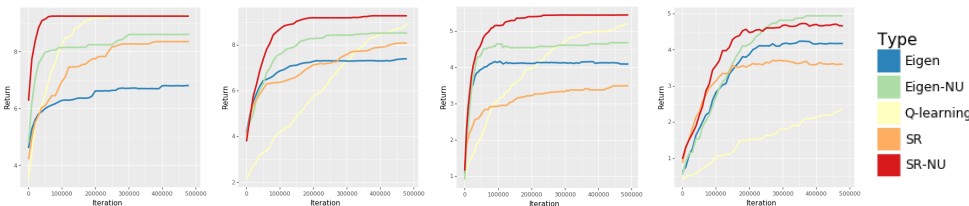

Figure 13: Performance plots

## J DENSITY PLOT FOR OPTIONS

In order to understand the necessity of the non-uniform exploration strategy, we look at the visitation count of various states when exploring with different policies (see Figure 14. Using only the actions result in low visitation counts at far-away states. However, the non-uniform strategy results in well distributed visitation counts across the state space.

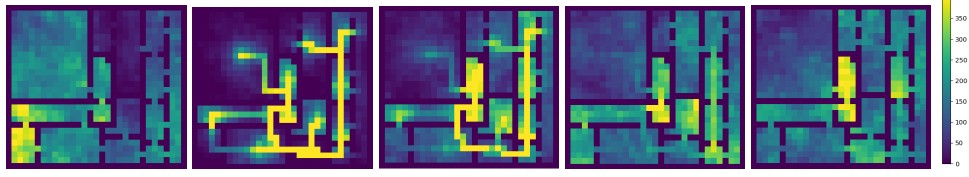

Figure 14: The visitation count (density) plot under different ratios for the non-uniform exploration policy. From left to right, the ratios are 1:0, 1:1, 1:19, 1:499 and 1:999

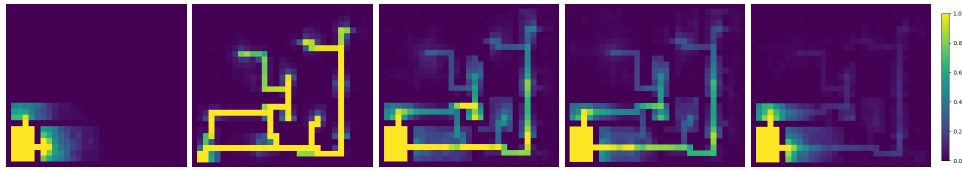

Figure 15: The visitation count (density) plot under different ratios for the non-uniform exploration policy in the finite horizon setting. From left to right, the ratios are 1:0, 1:19, 1:74 1:99 and 1:499

A possible optimization would be to explore in proportion to the size of the cluster, after selecting the option corresponding to that cluster. This pattern in visitation counts is indicative of the fact that the non-uniform exploration strategy is likely to be more effective. The policy is executed for $10^4$ primitive actions in the environment. Another point to note is that a higher ratio seems to perform less effectively. Hence, the options ensure that a key sequence of actions are performed to navigate to a different region in state space, which is consequently explored only if primitive actions are

taken. Sampling options in a frequent manner results in the agent spending a large fraction of its time near the termination states of these options.

We also plot the density maps for the finite horizon case (Figure 15. The agent is allowed to explore the environment for only 100 steps after which it is reset back to its start state ([1,1] in this experiment). The plots are visitation counts of the agent (for 100 primitive actions in the environment) averaged over 1000 runs.

## K INTRA-OPTION POLICY VISUALIZATION

The intra-option policies are visualized for both eigen-options (Figure 16 and SR-options (Figure 17). The green states are states, where the option terminates in a deterministic fashion. SR-options are run for 4 sub-goals while eigen-options uses the first 2 eigen vectors (4 options considering positive and negative version of each eigen-vectors to obtain eigen-options).

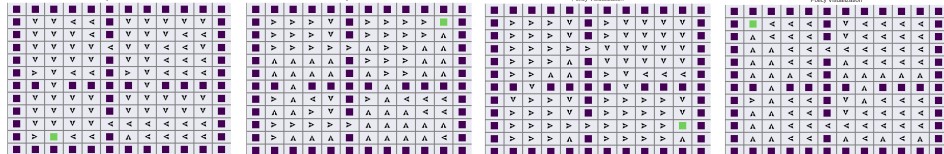

Figure 16: Eigen-option Intra-option policies

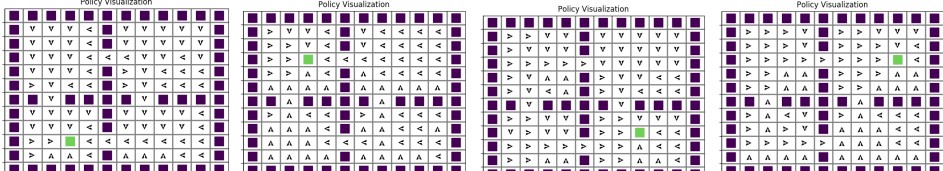

Figure 17: SR-option Intra-option policies

## L VARYING NUMBER OF OPTIONS

This plots (Figure 19) qualitatively demonstrate that the pre-determined number of options, changes the nature of the finally achieved sub-goals. The sub-goals are well separated and not redundant in nature. The method is hence expected to be fairly robust to the number of sub-goals. Figure

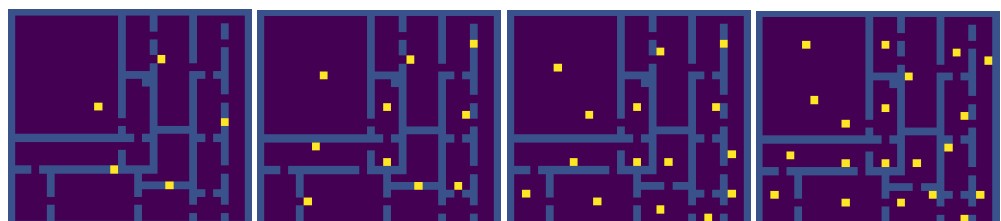

Figure 18: The sub-goals, when options are varied

19 shows the performance of SR-options, with a varying number of sub-goals. A small number of sub-goals may not easily reach all parts of the state space and a large number of sub-goals will result in poorer exploration (since there are more options to choose from). Hence, we see that 15 sub-goals are optimal for this problem. The plots are for the same grid-world as that in Figure 18. The setup is identical to that used in the earlier evaluations. After learning the options, 100 different scenarios of start and end states were used. For each scenario the training was done for $5 \times 10^5$ steps (primitive actions) in the environment and the evaluation of return was done 200 times in equally spaced intervals. Figure 19 is a result of averaging the return curves for the 100 different scenarios of start and end states.

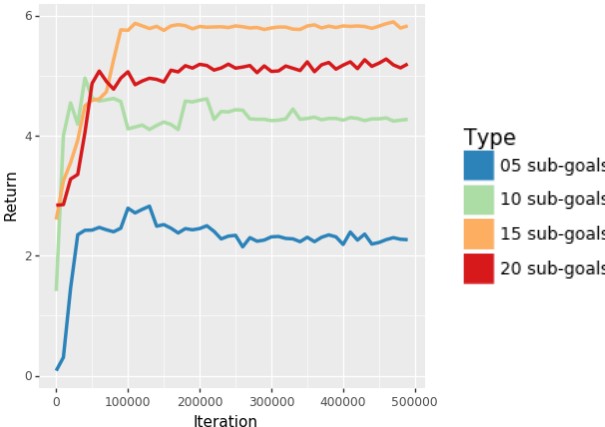

Figure 19: The performance curves for varying number of sub-goals

## M VARYING TRADE-OFF BETWEEN ACTIONS AND OPTIONS IN EXPLORATION

This section varies the ratio of sampling action and options in the non-uniform exploration scheme. The performance is optimal for a large exploration ratio of 1:999 which is not necessarily surprising, since Figure 14 indicates that this scheme covers the state-space most effectively. The experimental conditions are same as that described in Appendix L (similar to main experiments for evaluating different methods).

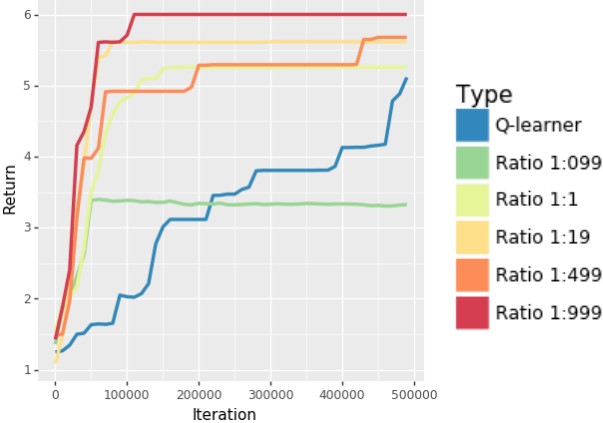

Figure 20: Performance curves for different ratios of sampling action and option in non-uniform scheme

## N CHOICE OF POLICY FOR LEARNING SR

We varied the policies used to collect the Successor representations to visualize the nature of the sub-goals. This policy can be understood as a prior over the state space, when deciding sub-goals. Figure 21 highlights that a policy biased towards a region in state space (say top room in 2-room domain) results in more sub-goals present in that region in state space. This is a result of the fact that the SRs are more detailed in these regions resulting in a fine-grained clustering of that region. Hence, if we have a policy that operates on an "important" part of the state space, SR-options automatically defines sub-goals in only these regions in state space.

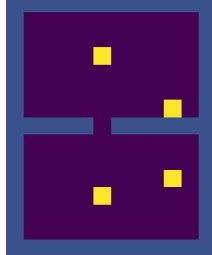 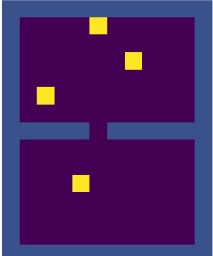

Figure 21: The sub-goals when the following policies for [No-op, Left, Right, Up, Down] are used. Left Image:(0, 0.3, 0.3, 0.3, 0.1) and Right Image:(0, 0.1, 0.3, 0.3, 0.3)

## O    DENSE NATURE OF INTRINSIC REWARD

While the Successor representations may not have a large magnitude in all parts of the state space, the same is not required to quickly learn a policy. The cardinal requirement of a good intrinsic reward is to ensure that the largest reward, points the agent in the right direction, which is the case here. For example, in Figure 22, the difference in values correspond to the reward, which although small guides the agent in the right direction. This is a good enough signal to determine the right action to take from every state with ease.

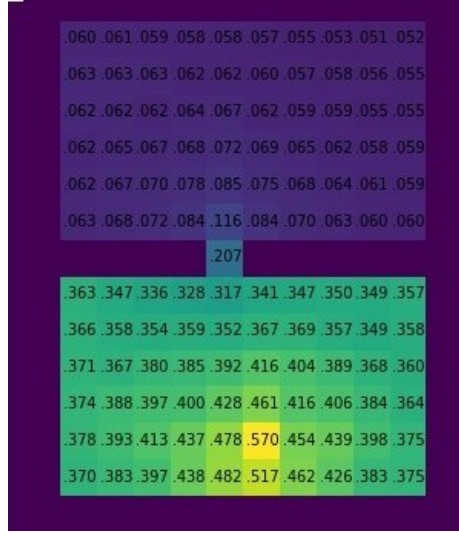

Figure 22: The Successor representation in the 2-room domain, visualized with values.

