# OpenReview forum: "Successor Options : An Option Discovery Algorithm for Reinforcement Learning"
_ICLR.cc/2019/Conference_

### Official Review · AnonReviewer1 · 2018-11-02
**Interesting idea and direction, but both the method and the derived insights need more work and understanding.**

**Rating:** 4
**Confidence:** 5

**Review:**


Summary: This paper tries to tackle the option discovery problem, by building on recent work on successor representation and eigenoptions. Although this is an extreme important problem, I feel the paper fails to deliver on its promise. The authors propose a way of clustering states via their SR/SF representation and they argue that this would lead to the discovery of subgoals that are fundamentally different from the popular choices in literature, like bottleneck states. They argue that this discovery procedure would lead to states “better for exploration”, “provide greater accessibility to a larger number of states”. Both of which sound promising, but I felt the actual evaluation fails to show or even assess either of these rigorously. Overall, after going through the paper, it is not clear what are the properties of these discovered subgoal states and why they would be better for exploration and/or control.

Clarity: Can be improved significantly! It requires several reads to get some of the important details. See detailed comments.

Originality and Significance: Very limited, at least in this version. The quantitative, and in some cases qualitative, evaluation lacks considerably. The comparison with the, probably most related, method (eigenoption) yield some slight improvement. But unfortunately, I was not conceived that this grain of empirical evidence would transfer to other scenarios. I can’t see why that would that be the case, or in which scenarios this might happen. At least those insights seem to be missing from the discussion of the results.


Detailed comments and questions:

1) Section 3.1: Latent Learning. There are a couple of design choices here that should have been more well explained or motivated:
i) The SR were built under the uniformly random policy. This is a design choice that might work well for gridworld/navigation type of domains but there are MDP where the evaluation under this particular policy can yield uninformative evaluations. Nevertheless this is an interesting choice that I think deserved more discussion, especially the connection to previous work on proto-value functions and eigenoptions. For instance, if both of these representations -- eigenoptions and the proposed successor option model -- aim to represent the SR under the uniform policy, why does know do (slightly) better than the other? Moreover, how would these compare under a different policy (non-uniform).
ii) The choice of reward. The notation is a bit confusing here, as it’s somewhat inconsistent with the definitions (2-4). Also, more generally, It is not clear throughout if we are using the discounted or undiscounted version of SR/SFs -- (2-3) introduce the discounted version, (4) seems to be undiscounted. Not clear if (5) refers to the discounted or undiscounted version. Nevertheless, I am guessing this was meant as a shaping reward, thus \gamma=1 for (5). But if that’s the case, according to eq. (2), most of the time I would expect \psi_s(s_{t+1}) and \psi_s(s_{t}) to have the same value. Could you explain why that is not true (at least in your examples)?
iii) Termination set: Q(s,a)<=0. This again seems a bit of an arbitrary choice and it’s not clear which reward this value function takes into account.

2) Figure 2: The first 2 figures representing the SRs for the two room domain: the values for one of the rooms seems to be zero, although one would expect a smoother transition around the ‘doorway’, otherwise the shaping won’t point in the right direction for progression. Again, this might suggest that more informative(control) policy might give you more signal.

3) Section 3.2: ‘The policy used for learning the SR is augmented with the previously learnt options‘. Can you be more precise about how this is done? Which options used? How many of them? And in which way are they used? This seems like a very important detail. Also is this augmented policy used only for exploration?

4) SRmin < \sum_{s’} ψ(s, :) < SRmax. Is this meant to be an expectation over all reachable next states or all states in the environment? How is this determined or translated in a non-tabular setting. Not sure why this is a proxy to how ‘developed’ this learning problem or approximation is. Can you please expand on your intuition here?

5) Section 3.3. The reward definition seems to represent how much the progress between \phi(s_t+1) - \phi(s) aligns with the direction of the goal. This is very reminest of FuN [2] -- probably a connect worth mentioning and exploring.

6) Figure 4: Can you explain what rho is? It seems to be an intermediate representation for shared representation \phi. Where is this used?

7) Experiments:
“a uniformly random policy among the options and actions (typically used in exploration) will result in the agent spending a large fraction of it’s time near these sub-goals”. Surely this is closely linked to the termination condition of the option and the option policy. How is this assessed?

“in order to effectively explore the environment using the exploration policy, it is important to sample actions and options non-uniformly”. It would be good to include such a comparison, or give a reason why this is the case. It’s also not clear how many of the options we are considering in this policy and how extensive their horizons will be. This comes back to the termination condition in Section 3.1 which could use an interpretation.

“In all our experiments, we fix the ratio of sampling an option to an action as 1:19.” This seems to be somewhat contradictory to the assumption that primitive actions are not enough to explore effectively this environment.

Figure 8. I think this experiment could use some a lot more details. Also it would be good to guide the reader through the t-SNE plot in Figure 8a. What’s the observed pattern? How does this compare to the eigenoption counterpart.

8) General comment on the experiments: There seems to be several stages in learning, with non-trivial dependencies. I think the exposition would improve a lot if you were more explicit about these: for instance, if the representation continually refined throughout the process; when the new cluster centers are inferred are the option policies learnt from scratch? Or do they build on the previous ones? Does this procedure converge -- aka do the clusters stabilize?

9) Quantitative performance evaluation was done only for the gridworld scenarios and felt somewhat weak. The proposed tasks (navigation to a goal location) is exactly what SFs are trained to approximate. No composition of (sub)tasks, nor tradeoff-s of goals were studied [1,3] -- although they seem natural scenario of option planning and have been studied in previous SFs work. Moreover, if the SFs are built properly, in these gridworlds acting greedily with respect to the SFs (under the uniformly random policy) should be enough to get you to the goal. Also, probably this should be a baseline to begin with.

References:
[1] Andre Barreto, Will Dabney, Remi Munos, Jonathan J Hunt, Tom Schaul, Hado P van Hasselt, and ´ David Silver. Successor features for transfer in reinforcement learning. In Advances in Neural Information Processing Systems, pp. 4055–4065, 2017.

[2] Vezhnevets, A.S., Osindero, S., Schaul, T., Heess, N., Jaderberg, M., Silver, D. and Kavukcuoglu, K., 2017, July. FeUdal Networks for Hierarchical Reinforcement Learning. In International Conference on Machine Learning (pp. 3540-3549).

[3] Barreto, A., Borsa, D., Quan, J., Schaul, T., Silver, D., Hessel, M., Mankowitz, D., Zidek, A. and Munos, R., 2018, July. Transfer in deep reinforcement learning using successor features and generalised policy improvement. In International Conference on Machine Learning (pp. 510-519).

---

> ### Author Response · Authors · 2018-11-21
> **Response (1/2)**
>
> Thank you for reviewing our work and for providing valuable feedback. We attempt to address some of the questions raised below:
>
> --Significance:
> What environments does this work on? This should work with any environment with complicated reward structures (conjunction of multiple rewards for example  or moving back and forth between rooms), provided that the environment has reversible transitions. The reversibility enforces that the SR-options can be learnt using our proposed intrinsic reward. We would also argue that questioning the applicability of SR-options is tantamount to questioning the usability of Eigen-options or bottleneck based options. Another area of applicability is the finite horizon case where the episode ends after 100 steps. Note that techniques such as Eigen-options fail in such a case.
>
> -- Properties of the discovered subgoal states:
> The aim of SR-options is to navigate to states that are “landmark states” or representatives of regions. Formally, this would translate to a state that has the largest value of  $\phi(s)  \cdot \Psi(cluster-center)$. Figure 5 effectively conveys the qualitative nature of SR-Options. The state coloured in yellow (each state corresponds to the termination state for a different option) indicates that each option leads the agent to a different part of state space.
>
> -- Improvement in performance over Eigen-options:
>
> We believe that we vastly improve over eigen-options (magnitude of improvement depends on the evaluation metric, the reward and the environment size). It is hence not trivial to label the improvement as mild/vast. To clarify this point further, we have added Appendix J (See Figure-13) which highlights this fact more clearly and plots the training curves (Figure 6 reports the area under these curves). We report scores at 200 different intervals and a difference of around 30-50 in the AUC metric is in-fact a vast improvement in performance since this implies we learn 50000 steps earlier than all other methods.
>
>
> 1) Choice of latent learning
>
> 1a) Uniform random policy: The choice of the policy is something we did not discuss due to a constraint on space. We have added an Appendix N that delineates some details on the same. The policy used to form an SR is effectively a prior over the state space, determining which parts of the state space are relevant. Hence, a policy that results in the agent spending more time in a certain region in state space, will result in a larger number of sub-goals in that region. This is clearly demonstrated in Figure 21. Hence a uniformly random policy is a suitable choice for our experiments since we do not prefer any part of the state space over the other.
>
> 1b) Choice of reward and termination condition: The primary goal of each option is to reach the state $s$ where $SR(cluster-center) . \phi(s)$ is maximized. Hence in order to encourage the agent to maximze this values, we define the pseudo reward $r(s_t, s_{t+1} = SR(cluster-center). (\phi(s_{t+1}) - \phi(s_t))$ (No gamma here). In a grid-world $\phi(s)$ is a one-hot vector and hence this would correspond to terminating the option when a state with the highest visitation count is reached (hence our claim of landmark sub-goals). Once we reach the state with maximum value of $SR(cluster-center). \phi(s)$, we want to terminate the options and hence Q(s,a) <=0 (Q-value under proposed intrinsic reward) which is satisfied for this state, is a suitable condition. Moving to any other state from this state will result in a negative reward.
>
> 1c)To clarify, Equation 4 is a typo and should include a discount factor \gamma. Thank you for pointing out the same.
>
> 1d) Identical SR value of neighbouring states: For the case of $\gamma=1$, Equation 2 effectively equates the SR, to a vector encoding the visitation counts of different states. So while neighbouring states have similar counts (smoothness expected) it is understandable that they do not have the exact same visitation count. Hence \psi_s(s_{t+1}) and \psi_s(s_{t}) are unlikely to have the same value when an expectation over visitation count is applied.
>
> 2) SR and smoother transition:
> The SRs do not have very smooth transitions across boundaries and this is further indicative of the fact that a uniformly random policy has a low probability of transitioning from one well-connected region to another (hence we need options to do the same). However, even if the magnitudes are very small, this does not necessarily mean that the agent lacks sufficient information.  The shaping always points in the right direction even if the shaping rewards have small magnitudes. This small magnitude suffices for the agent to determine the suitable action from each state. Figure 22 (Appendix O) visualizes the SR values in a more lucid fashion.

---

> ### Author Response · Authors · 2018-11-21
> **Response (2/2)**
>
> 3) SR Augmented with previously learnt options:
> The incremental SR-options procedure alternates between building the SR and clustering the learnt SR to generate options.
> Which option used? : The options generated from SR-options derived from the current SR. The SR is constantly modified and made more accurate and the options change based on the current values of the SR.
> How many of them?: This is the hyper-parameter which is decided beforehand like in the SR-options algorithm. This remains fixed throughout training.
> What way are they used?: They are used to learn the SR using a TD-based update rule (Equation 3). They are used in conjunction with the actions as in any other options framework. As mentioned in the paper, the SR is updated only when a primitive action is executed (not when executing options).
>
> 4) Why is L1 norm a proxy for how developed SR is:
> Machado et al. [1] demonstrate that the L1 norm of the SR can be used as a proxy for the count in the grid-world and function approximator settings. The intuition for the same is that the L1-norm of the SR (for the grid-world case) is exactly equal to 1/(1-gamma) which is shown in Appendix E. As a result, the L1-norm of the SR starts from zero and increases with every update to this constant. Hence the magnitude of the SR serves as a good proxy for the visitation count of that state.
>
> 5)Relation to FuN:
> This seems like an interesting parallel between the two methods. While FuN has a manager and a sub-manager with the manager directing the sub-manager towards a region in state-space, SR-options also has controllers operating at two levels in a hierarchy with the higher level directing the lower level controller to certain sub-goals but instead using a pseudo reward. We will expand on the similarities in a revised version of the paper.
>
> 6) Rho in figure 4:
> Rho in Figure-4 is the layer that is reconstructed like an auto-encoder (auxiliary loss), present in the top-most head of the network (as indicated in the top right of Figure 4). This is necessary to ensure that the built SR does not learn a trivial solution like the null-vector.
>
> 7a) Figure 8a t-SNE plot:
> The t-SNE plot demonstrates that the SR constructed by the function approximator is able to spatially segregate the states like in Figure 9. Hence the learnt SR is able to infer the graphical structure even for high dimensional spaces.
>
> 7b) Utility of NU scheme:
> “This seems to be somewhat contradictory to the assumption that primitive actions are not enough to explore effectively this environment”.
> While primitive actions on their own may not suffice, this does contradict the fact that we may need to use actions more frequently than options. Options lead us to new regions in the state space where primitive actions have a hard time reaching. However, primitive actions are still required to explore the region that the option has navigated to.
>
> The reasoning for the same is that an option terminates at a specific state. Every time that option is called, it terminates at a state ‘s’ with the highest visitation count. Hence, the agent will spend a large fraction of its time around that state. This intuition is justified in Appendix J (Figure 14 and Figure 15) which demonstrates that the non-uniform exploration scheme is more beneficial.
>
> 8) More details on the incremental approach:
> We will definitely improve the exposition and make the final version more clear to the reader. When the new clusters are built, the options policies are learnt from scratch since it is not trivial to map the old set of options to the new ones we intend to learn. The clusters stabilize in the end, because of the fact that we sample a collection of states based on their L1 norm. However, when the SR of all states are reasonably developed, the L1 norm of all states is expected to be identical (See Appendix E) and hence we have a random sample of states in the end. This would be equivalent to performing the original SR-options procedure.
>
> 9) Suggested baseline approach:
> Acting greedily wrt to the SRs should be sufficient to reach the goal for the grid-worlds we worked with. However, this isn’t the case universally. We do not add this baseline since this method is not generic enough and will fail when there are two rewards present, for instance. Using the SR will result in following the Q-value function of the uniformly random policy which clearly need not be optimal for a large number of problems.
>
> We hope the above response clarifies most of the concerns and queries raised by the reviewer and we would be happy to clarify more details. We hope the reviewer takes the above rebuttal into account and modifies the score appropriately.
>
> [1 ] Machado, Marlos C., Marc G. Bellemare, and Michael Bowling. "Count-based exploration with the successor representation." arXiv preprint arXiv:1807.11622 (2018).

---

### Official Review · AnonReviewer3 · 2018-11-03
**A sound technique, but incremental comparing to previous methods learning eigenoption and learning bottleneck states based on SRs.**

**Rating:** 6
**Confidence:** 4

**Review:**

This paper studies of the problem of HRL. It proposed an option discovery algorithm based on successor representations (SR). It considers both tabular and function approximation settings. The main idea is to first learn SR representation, and then based on the leant SR representation to clusters states using kmeans++ to identify subgoals. It iterate between this two steps to incrementally learn the SR options. The technique is sound, but incremental comparing to previous methods learning eigenoption and learning bottleneck states based on SRs.

Here are the comments for this manuscript:

How sensitive is the performance to the number of subgoals? How is the number of option determined?

It is unclear to me why the termination set of every option is the set of all states satisfying Q(s, a)\leq 0. Such a definition seems very adhoc.

Regarding the comparison to eigenoptions, arethey learned from SR as well? It seems that the cited paper for eigenoptions is based on the graph laplacian. If that is true, a more recent paper learns eigenoptions based on SR should be discussed and included for the comparison.

Machado et al., Eigenoption Discovery through the Deep Successor Representation, ICLR2018

---

> ### Author Response · Authors · 2018-11-21
> **Response**
>
> Firstly, thank you for spending your time, reviewing our work and for the valuable feedback. We hope to answer a few of the concerns and queries raised by the reviewer.
>
> 1) Incremental comparing to previous methods learning eigen-option and learning bottleneck states based on SRs.
>
> We believe that our work is significantly different from the Eigen-options [1] and Eigen-options based on Successor Representation [2]. Our primary contribution in this work is to provide an option discovery mechanism to landmark sub-goals, a type of sub-goal that hasn’t been considered in prior works. With regards to the work on Eigen-options with Successor Representations, Machado et al. [2] prove a theorem indicating that under certain assumptions, the options discovered by Eigen-options and Successor Representation based Eigen-options are identical in nature. Hence there isn’t any similarity to our work, in terms of the nature of options discovered, or the manner in which Successor Representations are used. We believe that the only commonality between the two papers is the problem of option discovery that both works attempt to address.
>
> 2) How sensitive is the performance to the number of subgoals? How is the number of options determined?
> We did not excessively tune the number of sub-goals and chose the number to be close to the number of rooms in that environment. Figure 19 in Appendix “L” varies the number of sub-goals for the 3rd environment (3rd Figure from left in Figure 3). We don’t observe a significant variance in performance and believe our method is fairly robust to this hyper-parameter and Figure 18 qualitatively explains the same since the locations of the sub-goals adapt to the hyperparameter, the total number of sub-goals.
>
> 3) It is unclear to me why the termination set of every option is the set of all states satisfying Q(s, a)\leq 0. Such a definition seems very adhoc.
>
> The aim of each of each option from SR-options is to reach a state $s’$ such that $s’ = argmax_{k} \phi(k) \cdot SR(cluster-center)$. The reward function naturally encodes the same, by assigning a reward of r(s_i, s_j) = SR(goal) [ \phi(s_j) - \phi(s_i) ] i.e. we attempt to move to states with higher values of the component of SR. Q(s,a)<=0 for a point where $\phi(s’) \cdot. SR(cluster-center)$  is the highest possible value and hence the option is set to terminate here. Hence we use this condition. In the tabular case, $\phi(s’)$ is a one-hot vector. Hence $\phi(s’) \cdot. SR(cluster-center)$ would correspond the state which has highest visitation count starting from “cluster-center”. Hence Q(s,a)<=0 encodes the intuition that we terminate the option once we land in a state with the highest discounted visitation count (dictated by SR) or in other words, a “landmark state”. This is precisely the reason that we claim that SR-options naturally direct you towards landmark states.
>
> 4) Regarding the comparison to eigenoptions, are they learned from SR as well? It seems that the cited paper for eigenoptions is based on the graph laplacian. If that is true, a more recent paper learns eigenoptions based on SR should be discussed and included for the comparison.
>
> We learn the eigenoptions based on the graph laplacian and not based on the SR. However. Machado et al. [2] show that the eigen-vectors of the SR (uniformly random policy) and that of the graph Laplacian are identical. Hence the learnt options are exactly identical in nature (for both methods). Therefore, the more recent paper will result in no differences in terms of qualitative and quantitative evaluation. This also adds to the point that our work is fundamentally different from [2]. Although Machado et al. [2] work with the SR, the discovered options are still the results of using the eigen-vectors of the graph laplacian and don’t yield different options.
>
> We hope the above response clarifies some of the concerns and queries raised by the reviewer and we would be happy to further clarify any more details. We hope the reviewer takes the above rebuttal into account and modifies the score appropriately.
>
> [1] Machado, Marlos C., Marc G. Bellemare, and Michael Bowling. "A laplacian framework for option discovery in reinforcement learning." arXiv preprint arXiv:1703.00956 (2017).
> [2] Machado, Marlos C., et al. "Eigenoption Discovery through the Deep Successor Representation." arXiv preprint arXiv:1710.11089 (2017).

---

### Official Review · AnonReviewer4 · 2018-11-11
**Interesting ideas but experimental evaluation lacking**

**Rating:** 5
**Confidence:** 4

**Review:**

The authors propose a method based on successor representations to discover options in a reward-agnostic fashion.  The method suggests to accumulate a set of options by (1) collecting experience according to a random policy, (2) approximating successor representation of or states, (3) clustering the successor representations to yield “proto-typical” cluster centers, and (4) defining new options which are the result of policies learned to “climb” the successor representation of the proto-typical state.  The authors provide a few qualitative and quantitative evaluations of the discovered options.

I found the method for discovering options reasonable and interesting.  The authors largely motivate the method by appealing to intuition rather than mathematical theory, although I understand that many of the related works on option discovery also largely appeal to intuition.  The visualizations in Figure 5 (left) and Figure 7 are quite convincing.

My concerns focus on the remaining evaluations:

-- Figure 5 (right) is difficult to understand.  How exactly do you convert eigenoptions to sub-goals?  Is there a better way to visualize this?

-- The evaluation method in Figure 6 seems highly non-standard; the acronym AUC is usually not used in this setting.  Why not just plot the line plots themselves?

-- Figure 8 is very difficult to interpret.  For (a), what exactly is being plotted? SR of all tasks or just cluster centers?  What should the reader notice in this visualization?  For (b), I don’t understand the options presented.  Is there a better way to visualize the goal or option policy?

-- Overall, the evaluation is heavy on qualitative results (many of them on simple gridworld tasks) and light on quantitative results (the only quantitative result being on a simple gridworld which is poorly explained).  I would like to see more quantitative results showing the the discovered options actually help solve difficult tasks.

---

> ### Author Response · Authors · 2018-11-21
> **Response**
>
> Thank you for reviewing our work and for providing valuable feedback. We will attempt to answer all of the queries raised in this response.
>
> 1)Explaining Figure 5
>
> In eigen-options, the termination condition is deterministic. We consider a sub-goal to be those states, where the option terminates. Each eigen-option is tied to a specific colour, and the colours marked on the grid-world correspond the sub-goals of the respective options (states where eigen-option terminates). Note that each eigen-option can have multiple sub-goals or just a single sub-goal (at least one sub-goal guaranteed since every option terminates).  The plot illustrates that eigen-options often have overlapping sub-goals (See Figure 5, two images on the right), which implies that the various eigen-options terminate in nearby regions in state-space. In contrast, each SR-option is useful since it leads the agent to a region in state space, that no other option navigates to (See Figure 5, two images on the right).
>
> 2)The evaluation method in Figure 6 seems highly non-standard; the acronym AUC is usually not used in this setting. Why not just plot the line plots themselves?
>
> We wanted to use a single metric in order to compare the different methods (based on the speed of learning and quality of final policy, both of which are captured in the defined AUC). Although the AUC is not commonly used, it can be argued to be extremely similar to regret, commonly used in Bandit literature. We have also added the line-plots (See Appendix I, Figure 13) which more clearly highlights the improvement in performance as a result of using SR-options.
>
> 3) Interpreting Figure 8:
>
> Figure 8 plots the successor representation for a sample set of states in Deepmind-Lab. The plot indicates that the Successor representations are segregated into 3 distinct clusters, indicating that the SRs learn to segregate the states into 3 regions. We are looking at other methods to visualize the Deepmind-Lab options since an aerial view isn’t viable. The first option moves down along a corridor and the second option goes through the door, from 1 room to another.
>
> 4) Other environments:
>
> Since other option discovery methods are also applicable to solving the tasks we consider, we modify them so as to improve the task difficulty further. This is done by restricting each episode to a fixed horizon, thus not allowing random walks to aid in obtaining a well-represented SR and hence good SR-options. We argue that this is a reasonable setting as (i) Many complex tasks restrict access to parts of state space inherently when using random actions because of their local dynamics (consider Montezuma’s revenge where random exploration does not allow visiting all parts of the state space) and (ii) analysing over grid-worlds still allows us to gather useful insights on how better our method performs over prior techniques (this is difficult in the function approximation setting). Such a setting is precisely realized in the incremental SR-options case, where we show that directed exploration is possible even when most parts of the state space is initially inaccessible. The final set of options are better equipped in solving any reward based task as compared to the initial starting set. This is not possible when using other techniques such as eigen-options.
>
> The primary goal of these options is to provide the agents access to different parts of the state-space by performing a sequence of transitions that typically have a low probability. Hence, regardless of the reward structure (say conjunction of multiple rewards), these options are useful on tasks which have reversible transitions (since latent learning is valid in this situation). Tasks which require going back and forth between two regions in state space (say retrieving a key in another room to open door in one room) would also benefit from SR-Options. It is also easy to make the argument that SR-Options benefit cases where Eigen-options and Bottleneck options are useful in nature.
>
> We hope the above response clarifies most of the concerns raised by the reviewer and we would be happy to further clarify any more details. We hope the reviewer takes the above rebuttal into account and modifies the score appropriately.

---

> > ### Comment · AnonReviewer4 · 2018-11-22
> > **Additional Comments**
> >
> > Thank you for these clarifications, some of my comments are below - hopefully they can further improve the presentation of the paper.  However, I believe the paper still significantly lacks from a strong quantitative demonstration of the usefulness of successor options (or alternatively, more theoretical justification, although I think this is much harder to achieve).  While overall this is not a bad paper, in general and as my score indicates, I would lean towards rejecting.
> >
> > Figure 6 & 13
> > The plots here could use more description in the caption (and main text).  What do the four plots correspond to?  What exactly is the task/environment?
> >
> > "The plot indicates that the Successor representations are segregated into 3 distinct clusters"
> > I guess this is subjective, but the shown clusters do not seem especially clear; i.e., they seem quite noisy and of varying shapes.  I think alternative modes of visualization would help here.

---

> > > ### Author Response · Authors · 2018-11-26
> > > **Response to Comments**
> > >
> > > Figure 6 and 13:
> > > The 4 plots correspond to the 4 grid-worlds in Figure 3 (More details given in Appendix B).  We see that this can be confusing to a reader and will make this fact more explicit in the paper. We will also expand on the captions to clarify the same.
> > >
> > > The comments definitely point us in directions along which we can improve the paper readability and we thank you for the same and will attempt to incorporate the same.

---

### Official Review · AnonReviewer2 · 2018-11-27
**This could be an interesting paper but currently requires a lot of experimental and writing improvements**

**Rating:** 4
**Confidence:** 5

**Review:**

The paper proposes to use successor features for the purpose of option discovery.  The idea is to start by constructing successor features based on a random policy, cluster them to discover subgoals, learn options that reach these subgoals, then iterate this process.  This could be an interesting proposal, but there are several conceptual problems with the paper, and then many more minor issues, which put it below the threshold at the moment.

Bigger comments:
1. The reward used to train the options (eq 5) could be either positive or negative.  Hence, it is not clear how or why this is related to getting options that go to a goal.
2. Computing SRs only for a random policy seems like it will waste potentially a lot of data. Why not do off-policy learning of the SR while performing  the option?
3. The candidate states formula seems very heuristic. It does not favour reaching many places necessarily (eg going to one next state would give a 1/(1-gamma) SR value)
4. Fig 5 is very confusing. There are large regions of all subgoals and then subgoals that are very spread out.  If so many subgoals are close by, why would an agent explore? It could  just jump randomly in that region for a while. It would have been useful to plot the trajectory distribution of the agent when using the learned options to see what exactly the agent is doing
5. There are some hacks that detract from the clarity of the results and the merits of the proposed method. For example, options are supposed to be good for exploration, so sampling them less would defeat the purpose of constructing them, but that is exactly what the authors end up doing. This is very strange and seems like a hack. Similarly, the use of auxiliary tasks makes it unclear what is the relative merit of the proposed method. It would have been very useful to avoid using all these bells and whistles and stick as closely as possible to the stated idea.
6. The experiments need to be described much better. For example, in the grid worlds are action effects deterministic or stochastic? Are start state and goal state drawn at random but maintained fixed across the learning, or each run has a different pair? Are parameters optimized for each algorithm?  In the plots for the DM Lab experiments, what are we looking at? Policies? End states? How do options compare to Q-learning only in this case? Do you still do the non-unit exploration? The network used seems gigantic, was this optimized or was  this the first choice that came to mind? Would this not overfit? What is the nonlinearity?

Small comments:
- This synergy enables the rapid learning Successor representations by improving sample efficiency.
- Argmax a’ before eq 1
- Inconsistent notation for the transition probability p
- Eq 3 and 4 are incorrect (you seem to be one-off in the feature vectors used)
- Figure 2 is unclear, it requires more explanation
- Eq 6 does not correspond to eq 5
- In Fig 6 are the 4 panes corresponding top the 4 envs.? Please explain. Also this figure needs error bars
- It would be useful to plot not just AUC, but actual learning curves, in order to see their shape (eg rising faster and asymptoting lower may give a better AUC).
- Does primitive Q-learning get the same number of time steps as *all* stages of the proposed algorithm? If not, it is not a fair comparison
- It would be nice to also have quantitative results corresponding to the experiments in Fig 7.

---

> ### Author Response · Authors · 2018-12-03
> **Response (2/2)**
>
> 6a. Experiment description
> Grid-worlds have always been deterministic and most works build on this assumption. Appendix B furnishes details on the environment.
>
> 6b. Are parameters optimized for each algorithm?
> We used the same set of hyper-parameters for all the 4 grid-worlds and hence we have not excessively tuned each value, indicating the robustness of the approach. Appendix E, Appendix L and Appendix M discuss some of these choices.
>
> 6c. Deepmind Lab experiments: The network used seems gigantic, was this optimized or was this the first choice that came to mind? Would this not overfit? What is the nonlinearity?
>
> Since our work focuses on option-discovery, we attempt to look at learnt options in the Deepmind Lab domain. Hence our only method for comparison is to look at other option discovery frameworks. Our network is in no way gigantic with respect to any Deep reinforcement learning work. DQN (Mnih et al[3]), uses a similar sized network for Atari. Our network has a couple more hidden layers owing to the fact that the RGB input for the Deepmind-lab task is almost 4 times bigger than the input from the Atari-2600 suite of games. Non-linearity is essential in generalizing well across high-dimensional images.
> With regards to the overfitting, Deep learning models tend to be heavily over-parameterized and rarely display any signs of overfitting. Furthermore, the concept of overfitting is still relatively undefined in Deep reinforcement learning since we lack clear definitions of train/test error.
>
>
> Small comments:
>
> 1. This synergy enables the rapid learning Successor representations by improving sample efficiency. AND Argmax a’ before eq 1
> A: The current version seems correct. Please do let us know if otherwise.
>
> 2. Inconsistent notation for the transition probability p
> Thank you for pointing this out. We will correct the same
>
> 3. Eq 3 and 4 are incorrect (you seem to be one-off in the feature vectors used)
> We believe the current version is indeed correct. Can you kindly clarify what you mean by “ you are one-off in the feature vectors used”
>
> 4. Figure 2:
> A: In particular, which aspects are unclear about this FIgure. The SR vector is for the two environments (from left) in Figure 3. The SR vector is plotted on the state space. This image highlights the fact that the reward function that we design ascends the Successor representation vector and indicates how the termination condition of an option looks like.
>
> 5. Eq 6 does not correspond to eq 5
> A: Equation 6 is the generic form of the reward function we propose for latent learning. For the case of the grid-world, the \phi(s_t) is a one-hot vector in which case, Equation 6 devolves into Equation 5.
>
> 6. Fig 6
> A: Indeed, the 4 figures in the top-pane correspond to the 4 bar-graphs (in the same order). Another reviewer has raised this same concern which we will clarify by modifying the appropriate caption/section. Thank you for pointing this out.
>
> 7. Plot of AUC
> A: We have plotted the learning curves in Appendix I for clarity.
>
> 8. Does primitive Q-learning get the same number of time steps as *all* stages of the proposed algorithm?
> A: This is not the case in the current setup because the Option-discovery is done only once, following which the learnt options are used to solve tasks with different start and end states. Since there are a large number of such configurations, the computation to learn the options is negligible in comparison. Hence, we do not account for the cost of Option discovery. Furthermore, it is unclear how one would do so if the options address multiple MDPs/tasks.
>
>
> [3]  Mnih, Volodymyr, et al. "Human-level control through deep reinforcement learning." Nature 518.7540 (2015): 529.

---

> ### Author Response · Authors · 2018-12-03
> **Response (1/2)**
>
> Thank you for going through our work and providing valuable feedback. We hopefully address the concerns and questions raised in this response and we would be happy to expand on any point unclear in this response.
>
> 1. The reward used to train the options:
>  One way to understand the proposed reward function is to look at Figure 22 (Appendix O). This reward function corresponds to one single option and dictates the intra-option policy for the same. The difference between the values printed on the individual states corresponds to the reward for a transition between the two states. Hence the designed reward function ensures that we reach a state with the largest value of $\phi(s) . |Psi(Sub-goal)$
>
> 2. Off-policy SR:
> Learning the SR in an off-policy manner is extremely hard and we are unaware of any such formulations. This is primarily because it is very difficult to relate the discounted visitation counts of one policy to that of another.
>
> 3. The candidate states formula seems very heuristic.
> We have a very good reason for the choice of the candidate states formula. The candidate states are those states which have a moderately developed SR. The formula uses the L1 norm which encodes the visitation count of a particular state (See [1]), hence encoding the degree to which the SR of a state is developed. The clustering from the SR-options ensures that the chosen sub-goals are sufficiently spread apart. Can you kindly clarify the comment: “going to one next state would give a 1/(1-gamma) SR value”. The maximum magnitude of the SR is 1/(1-gamma) so the statement isn’t too clear to us.
>
> 4. Fig 5:
> As stated in the image caption, the images on the left are for the Successor Options and the images on the right are for Eigen-options (the method we compare to). The coloured states (yellow to light blue) are the termination states of all options. This clearly emphasizes the point that the Eigen-options have nearby sub-goals. With regards to the statement (If so many subgoals are close by, why would an agent explore?), this is precisely the problem with the Eigen-options framework, and this is the basis on which we claim that Successor options will exhibit better empirical performance. With regards to the trajectory distribution, Appendix J addresses the same.
>
> 5. Experimental Evaluation:
> The reviewer claims that we use tricks to detract from the clarity of the paper, an assessment we politely disagree with.
>
> 5a. “Auxiliary tasks make relative merit unclear:
> Can the reviewer kindly clarify which auxiliary tasks are being referred to here? The only auxiliary loss we use is the image reconstruction loss (auto-encoder loss) which ensures that the Successor Features do not learn the null vector. This is a very commonly used loss in Successor Representation based papers ( See [2] which justify the usage of the same). This loss is very much part of our described framework and does not in an any form, hide the merit of the reported results
>
> 5b. Sampling options less would defeat the purpose of constructing them:
> We have demonstrated evaluations for Q-learning (only actions), and SR-options with uniform and non-uniform exploration and hence we do not “detract the clarity of the results” using hacks. We have attempted to be as transparent as possible in the reported results and we believe we stick to the method described. There are two points we would like to make 1) The primary focus of this work is on Option-discovery and hence we do not delve deeply into areas related to learning with options. There are a plethora of possibilities worth exploring and this does not detract from the utility of the options themselves. 2) Sampling the option less does not mean that there is no utility in constructing them. By that logic, one should expect a naive Q-learner to have the best performance (which is clearly not the case). So why is this scheme useful? Options are used to navigate to key parts of the state space (that primitive actions are unable to), while actions are used to explore the newly discovered region in state space. This is analogous to using an airplane to travel to a new city (using an option to land in new parts of state space), following which one explores the city on foot (each step is a primitive action). Hence the options are still essential, even if they are sampled infrequently since they perform a sequence of transition that has a negligible probability of happening. Sampling the options frequently would translate to jumping between regions in state space without exploring any one region. See Appendix J for more details
>
> [1 ] Machado, Marlos C., Marc G. Bellemare, and Michael Bowling. "Count-based exploration with the successor representation." arXiv preprint arXiv:1807.11622 (2018).
>
> [2] Kulkarni, Tejas D., et al. "Deep successor reinforcement learning." arXiv preprint arXiv:1606.02396 (2016).

---

### Public Comment · (anonymous) · 2018-10-15
**Intra-option policy heads**

Hi, one quick question, do you have one option policy head per option? And if so, then in the incremental successor options algorithm (3.2), when the cluster centers change on the go, how do you assign which option head should go to which cluster centers?
Regards.

---

> ### Author Response · Authors · 2018-10-16
> **Response**
>
> Thank you for spending time reading our work.
> The incremental successor options algorithm (3.2) addresses the tabular setting. Each option does have it's own option head (or more precisely it's own Q-value table). Hence, when the clusters change, the Q-value tables (and hence intra-option-policy) are all learnt from scratch for the newly assigned clusters. As a result, we do not have the problem of mapping options to appropriate option heads. This is because we do not re-use the value functions/policies from earlier options and the mapping is irrelevant.
>
> For the case of the function approximators, where each option has it's own option head, this problem needs to be addressed. This work does not deal with incremental successor options in a function approximation scenario. One possible solution would be to re-initialize all the option-policy head weights and consequently randomly assign the new options to option-policy heads. These option-policies are hence again learnt from scratch in a manner identical to the tabular setting.
>
> Hope this answers the question raised.

---

> > ### Public Comment · (anonymous) · 2018-10-17
> > **Thanks for the reply!**
> >
> > Thanks a lot for the reply, really interesting work and a really good read!

---

### Meta-Review · Area_Chair1 · 2018-12-15

**Confidence:** 4
**Recommendation:** Reject

**Metareview:**

Pros:
- simple, sensible subgoal discovery method
- strong inuitions, visualizations
- detailed rebuttal, 15 appendix sections

Cons:
- moderate novelty
- lack of ablations
- assessments don't back up all claims
- ill-justified/mismatching design decisions
- inefficiency due to relying on a random policy in the first phase

There is consensus among the reviewers that the paper is not quite good enough, and should be (borderline) rejected.